# Dementia and Sleep Disorders: The Effects of Drug Therapy in a Systematic Review

**DOI:** 10.3390/ijms26125654

**Published:** 2025-06-12

**Authors:** Luis Fernando Chavez-Mendoza, Alan O. Vázquez-Alvarez, Blanca Miriam Torres-Mendoza, Walter A. Trujillo-Rangel, Erandis D. Torres-Sánchez, Ismael Bracho-Valdés, Daniela L. C. Delgado-Lara

**Affiliations:** 1Department of Health Sciences, Institute of Biomedical Sciences, Autonomous University of Ciudad Juarez, Cd. Juarez 32310, Chihuahua, Mexico; chavezluis0305@gmail.com; 2School of Medicine and Health Sciences, Tecnológico de Monterrey, Guadalajara 45201, Jalisco, Mexico; md.alan.vazalv@gmail.com; 3Neurosciences Division, Western Biomedical Research Center, Mexican Institute of Social Security (IMSS), Guadalajara 44340, Jalisco, Mexico; bltorres1@hotmail.com; 4Department of Philosophical, Methodological and Instrumental Discipline, Health Sciences University Center, University of Guadalajara, Guadalajara 44340, Jalisco, Mexico; 5Department of Biomedical Sciences, Tonala University Center, University of Guadalajara, Tonala 45425, Jalisco, Mexico; walter.trujillo@edu.uag.mx; 6Dirección Académica de Aparatos y Sistemas II, Ciencias de la Salud, Universidad Autónoma de Guadalajara, Zapopan 45129, Jalisco, Mexico; 7Department of Medical and Life Sciences, University Center of la Cienega, University of Guadalajara, Ocotlan 47820, Jalisco, Mexico; erandis.torres@academicos.udg.mx; 8Dirección Académica de Aparatos y Sistemas I, Ciencias de la Salud, Universidad Autónoma de Guadalajara, Zapopan 45129, Jalisco, Mexico; ismael.bracho@edu.uag.mx; 9Departamento Académico de Formación Universitaria, Ciencias de la Salud, Universidad Autónoma de Guadalajara, Zapopan 45129, Jalisco, Mexico

**Keywords:** dementia, sleep disorders, pharmacotherapy, benzodiazepines, Z-drugs

## Abstract

Currently, approximately 40% of patients with dementia develop some form of sleep disorder. Benzodiazepines are widely prescribed but pose the risk of tolerance and cognitive decline; however, Z-drugs may offer safer alternatives. Therefore, this systematic review aimed to analyze the effect of benzodiazepines and Z-drugs on sleep disorders in patients with dementia. Two authors conducted a systematic search in PubMed, Scopus, Web of Science, Espistemonikos, and ACCESSSS for studies published between 2019 and 2024 using the MeSH terms “dementia”, “sleep disorders”, and “pharmacotherapy”. Randomized clinical trials comparing benzodiazepines, Z-drugs, or innovative medications with placebo or other drugs were included. Sleep and cognitive outcomes were assessed using validated instruments; the ROB-2 tool evaluated the risk of bias. The protocol was registered in “PROSPERO”. Three randomized clinical trials involving a total of 192 patients were included in the review. Zopiclone increased the main duration of nighttime sleep by 81 min, Zolpidem reduced nighttime awakenings by 21 min, and Eszopiclone improved sleep quality, benefited the progression of sleep architecture, and reduced mental symptoms such as fear and anxiety. Z-drugs show superior efficacy and safety over benzodiazepines, improving sleep and cognitive symptoms in dementia. Personalized treatment and further research across dementia subtypes are needed to optimize long-term outcomes.

## 1. Introduction

Dementia is a term that encompasses several diseases that affect memory, thinking, and the ability to perform daily activities. Currently, more than 55 million people worldwide have dementia, with over 60% living in low- and middle-income countries. The World Health Organization estimates that 10 million new cases occur each year [1]. In Mexico, the main subtypes of dementia include Alzheimer’s disease (7.8%), vascular dementia (4.3%), and mixed dementia (2.1%), with a higher prevalence in women (15.3%) compared to men (12.5%). Other important subtypes in the global context are Lewy body dementia (LBD) and frontotemporal dementia (FTD) [2].

Existing evidence demonstrates a strong association between dementia and sleep disorders—such as insomnia, obstructive sleep apnea (OSA), and rapid eye movement (REM) sleep behavior disorder (RBD)—and highlights the significant impact these disorders can have on the quality of life. Approximately 40% of individuals with dementia experience sleep disorders, and this percentage can rise depending on the type of dementia [3,4]. Poor sleep quality can be highly detrimental to memory, and at the same time, it is a factor in developing behaviors related to anxiety and depression. In patients with dementia, this secondary deterioration can compound problems [5].

Currently, non-pharmacological measures are considered the first-line treatment to solve sleep-related problems, with cognitive behavioral therapy and sleep hygiene being the mainstay of this alternative [6]. Other non-pharmacological approaches, such as light therapy for circadian rhythm disorders and CPAP for sleep apnea, may be appropriate depending on the specific sleep disturbance [7,8,9]. Despite this, multiple pharmacological alternatives complement non-pharmacological treatment measures. Among the pharmacological options, there are several drug classes used to manage sleep disorders in patients with dementia. These include sedatives and hypnotics (benzodiazepines and Z-drugs), orexin receptor antagonists (such as Suvorexant and Lemborexant), melatonin receptor agonists, sedative antidepressants (e.g., Doxepin and Amitriptyline), antipsychotics (such as Olanzapine), and antihistamines [10].

A thorough evaluation of the type of insomnia or sleep disorder, the underlying causes, and the degree to which it affects the patient’s daily life is necessary to choose the right medication. Additionally, factors such as the drug’s characteristics, risk–benefit profile, patient-specific considerations, potential side effects, and availability must all be considered. If an ideal hypnotic drug for treating insomnia existed, it would have a rapid onset of action, promote high-quality sleep throughout the night, leave no residual side effects the next day, and be affordable and widely available; however, such medication does not yet exist [11,12].

Benzodiazepines (BZDs) remain the most widely prescribed medications for sleep disorders in older adults [13]. They exert their hypnotic effect by enhancing the action of the inhibitory neurotransmitter of gamma-aminobutyric acid (GABA) at the GABA-A receptor and binding at the interface between the alpha and gamma subunits. Their sedative properties are mainly mediated through the alpha-1 subunit, while other effects, such as anxiolysis and muscle relaxation, involve alpha-2 and other subunits [14]. Despite their widespread use, concerns remain regarding their potential for tolerance, dependence, and adverse effects like cognitive impairment (especially in the elderly) [13,15]. Recently, non-benzodiazepine hypnotics, commonly known as Z-drugs, have been suggested as effective alternatives due to their similar mechanism of action at GABA-A receptors, but with greater selectivity for the alpha-1 subunit. This targeted binding may offer benefits for sleep quality with lower dependency risks, along with reduced adverse effects [15]. Furthermore, there needs to be a greater focus on sleep disorders as potential indicators of early-stage dementia [16].

Given these challenges, the question that arises is related to determining the best therapeutic approach for patients. The objective of this article is to analyze the effect of benzodiazepines and Z-drugs on sleep disorders in patients with dementia.

## 2. Materials and Methods

### 2.1. Search Strategy

A team member searched PubMed, Scopus, Web of Sciences, Epistemonikos, and ACCESSSS databases for studies published between 2019 and 2024 without automation tools. The following search algorithm was used: “(dementia) AND (sleep disorders) AND (Pharmacotherapy OR Drug Prescription) AND (Treatment Efficacy)”. The filters applied required that the full text be available and that the studies were no older than five years. The protocol was registered on the PROSPERO website (registration number CRD42024564935), the international prospective register of systematic reviews, and was accepted on July 16, 2024. The article was conducted following the Preferred Reporting Items for Systematic Reviews and Meta-analysis (PRISMA) guidelines.

### 2.2. Inclusion and Exclusion Criteria

The inclusion criteria were as follows: (1) Study type: clinical trials; (2) Population: people with dementia of any subtype with a sleep disorder; (3) Intervention: the use of benzodiazepines, Z-drugs, or any innovative medication; (4) Comparison: the use of a placebo or another benzodiazepine; (5) Outcome: assessing the efficacy of pharmacological therapy measured by objective sleep indicators such as sleep onset latency, efficiency, total duration, nighttime awakenings, and sleep architecture (NREM and REM); (6) Instruments used for sleep evaluation: actigraphy, polysomnography, and questionnaires such as the Pittsburgh Sleep Quality Index (PSQI) or the Epworth Sleepiness Scale; (7) Cognitive and behavioral evaluation: the Mini-Mental State Examination (MMSE) or the Montreal Cognitive Assessment (MoCA) were used; and (8) Language: unrestricted. The exclusion criteria were meta-analyses, systematic reviews, and protocols.

### 2.3. Data Extraction

Two individuals were responsible for extracting the information required for this systematic review, while a third investigator resolved any differences, disagreements, and uncertainties. The information collected from each included study was as follows: the first author’s name, year of publication, country, study type, age, sample size, gender, sleep disorder being treated, pharmacological therapy information, non-pharmacological therapy information, and therapy efficacy.

The review authors worked independently to assess the risk of bias using criteria described in the Cochrane Handbook for Systematic Reviews of Interventions to evaluate the quality of trials (ROB-2 tool). This set of criteria is based on evidence of associations between the potential overestimation of effects and the level of risk of bias in the article, which may arise from aspects such as sequence generation, allocation concealment, blinding, incomplete outcome data, selective reporting, and how these “domains” are reported.

In instances where the raters held differing opinions, the final rating was not a matter of individual preference but rather a result of a collaborative consensus. This involved another member of the review team ensuring that the risk of bias for each domain and the overall risk was assessed and categorized in a fair and balanced manner.
A.Low risk of bias: A plausible bias is unlikely to significantly alter the results (categorized as “No” in the risk of bias table).B.High risk of bias: A plausible bias that significantly weakens confidence in the results (categorized as “Yes” in the risk of bias table).C.Unclear risk of bias: A plausible bias that raises some doubt about the results (categorized as “Unclear” in the risk of bias table).

In cases where trial reports lacked adequate details regarding randomization or other study characteristics, we reached out to the authors for further clarification, ensuring thoroughness in our review process.

### 2.4. Data Synthesis

From the selected studies, the following variables associated with the efficacy of pharmacological therapy were identified and extracted through the following objective sleep indicators: sleep onset latency, efficiency, total duration, nocturnal awakenings, and sleep architecture (NREM and REM); the use of instruments for sleep evaluation included the following: actigraphy, polysomnography, and questionnaires such as the PSQI or the Epworth Sleepiness Scale; and cognitive and behavioral assessments performed included the following: the MMSE or the MoCA.

The data obtained from each article were presented with either a standard error or standard deviation (SD) according to its characteristics. If standard deviations were not reported and could not be calculated from the available data, the authors would be asked to provide them. In the absence of data from the authors, the SD would be calculated using the *p*-values and the sample size of the group(s) present in the individual studies.

## 3. Results

As shown in Figure 1, 164 articles were identified from the PubMed, Scopus, Web of Sciences, Epistemonikos, and ACCESSSS search. After applying the filters mentioned, 24 articles remained. Following the application of the exclusion criteria, 21 articles were removed, resulting in 3 articles, as presented in Table 1.

Based on the extracted studies, a risk of bias analysis was conducted, which is presented below in Figure 2.

Once the individualized evaluation of each study was complete, we integrated each result into a general assessment of all the obtained evidence and its corresponding risk of bias, as shown in Figure 3.

## 4. Discussion

Dementia is considered one of the most prevalent medical conditions affecting older adults today and is projected to continue increasing in relevance in the coming years [20]. Most available research focuses on Alzheimer’s disease, as it is the primary subtype of dementia. The bidirectional relationship between dementia and sleep disorders continues to be studied; however, the diagnosis and treatment of sleep disorders could contribute to protection against cognitive decline that might eventually evolve into dementia or the worsening of a pre-existing condition [21].

Dementia encompasses a progressive deterioration in cognitive function that is often preceded by subtle neuropathological changes many years before clinical onset [21,22]. Considering its frequent overlap with multiple comorbidities and the common use of several medications in this population, tailoring pharmacological strategies is essential.

In this review, we asked whether currently available pharmacological treatments are effective for managing sleep disorders in patients with dementia. We analyzed evidence from clinical trials to assess the effect of BZDs and Z-drugs on sleep disturbances in this population.

Currently, benzodiazepines remain the standard prescription for sleep disorders in older adults [23]. The National Ambulatory Medical Care Survey (NAMCS) recorded that the number of people prescribed at least one benzodiazepine increased from 27.6 million in 2003 to 62.3 million in 2015, with older adults being the most affected population [24]. Therefore, it is essential to consider their relationship with the progression of Alzheimer’s disease and related types of dementia.

In Latin America, there has been a higher prevalence of benzodiazepines among older adults, with diazepam being the most widely used drug for treating insomnia. Research in Mexico by Minaya et al. at the National Institute of Psychiatry Ramon de la Fuente Muñiz highlighted that benzodiazepine dependence was associated with more severe depression and anxiety symptoms and lower cognitive and psychosocial functioning [25]. Older adults are more sensitive to BZDs, which increases their risk of confusion, disorientation, falls, and the consequences that arise from them. This sensitivity is directly related to a significant systemic accumulation of these drugs and their metabolites. Benzodiazepines can primarily lead to toxicity in the central nervous system, motor impairment, sedation, and ataxia. These adverse effects increase their complexity in patients with existing dementia, where neurodegenerative deterioration predisposes them to cognitive and functional decline [26]. Fortunately, new pharmacological alternatives offer a promising outlook regarding acceptance and efficacy in treated patients [17,18,19].

It is important to remember that non-pharmacological treatment is currently the first line of intervention for patients with chronic sleep disorders, specifically cognitive behavioral therapy. These sleep disorders refer to difficulty maintaining sleep, sleep fragmentation, sleeping throughout the day, and the deterioration of a person’s quality of life [27]. However, the effectiveness of cognitive behavioral therapy is inconclusive, and it also entails a greater cost, effort, and time for the family and the patient.

The so-called “Z-Drugs” are non-benzodiazepine hypnotics commonly used in the treatment of insomnia. While Z-drugs may benefit patients with sleep disorders due to their shorter duration of action, their more rapid clearance than benzodiazepines, higher selectivity for the GABA-A receptor, lower risk of dependence, and fewer residual daytime effects, they also have adverse effects such as causing confusion, cravings, behavioral changes (sleep-walking, falls, and sleep-driving), which can result in serious injuries, and, in people with dementia, using higher doses has been associated with increased fracture and stroke risks, similar or greater to that experienced with higher doses of BZDs [28,29,30]. The influence of sex on drug metabolism is also important to consider. For example, Zolpidem has been shown to have a significantly longer half-life in women compared to men, potentially increasing the risk of residual sedation, daytime drowsiness, and associated functional impairments [31]. These findings underscore the necessity for careful patient selection and dose adjustments, especially in vulnerable populations like elderly patients with dementia.

Louzada and colleagues compared the efficacy of Zolpidem versus Zopiclone in patients with insomnia and Alzheimer’s disease. Patients treated with Zopiclone increased their total nighttime sleep duration by 81 min and experienced a 26 min reduction in wakefulness after sleep onset [18]. These superior benefits may be explained by the high non-selective affinity that patients have for BZD receptors, which means that Zopiclone binds to a wide range of receptors in the brain, potentially leading to more significant adverse effects in older individuals primarily due to the aging of their hepatic metabolism [32].

Based on this, a new question abruptly arises regarding the extent to which the benefits outweigh the risks in treating sleep disorders in patients with a pre-existing degenerative condition. Nevertheless, the benefits for such patients are latent, as reiterated in the study published by Shuyu Huo and colleagues, offering reassurance about the progress in dementia research and treatment [19]. The evaluation of the problem itself is a key point for treatment targeting [33].

Over a 21-month period, 96 patients diagnosed with Alzheimer’s disease and sleep disorders were studied, with Eszopiclone administered to the study group compared to Alprazolam for the control group. The results demonstrated not only an improvement in the secondary condition of patients’ sleep disorders but also a favorable evolution in some symptomatic repercussions of their illness. These symptomatic repercussions include cognitive decline, memory loss, and changes in behavior. There was a considerable improvement in sleep architecture, as well as positive changes regarding orientation, memory, calculations, and language skills [19].

While the two previous studies included a focus on the use of BZDs and Z-drugs, our criteria also allowed for the inclusion of innovative pharmacological interventions, given their potential to address specific sleep disorders in dementia. Ambra Stefani and colleagues analyzed the use of Nelotanserin for RBD conducted in patients with a prior diagnosis of LBD or Parkinson’s disease. Nelotanserin is a potent, selective inverse agonist of 5 HT, which has been previously investigated for its potential as a hypnotic agent. The results show no significant evidence of the effectiveness of treatment with Nelotanserin in these patients based on the parameters used to assess improvement. Moreover, these results are subject to multiple limitations, such as the small number of participants included, the lack of objective outcome measures, and the subjective measures used, such as questionnaires [17]. This indicates that innovation and pharmacological alternatives have a significant margin for improvement [34].

The strengths of our study include the absence of discrimination by language, an adequate search across different databases, and ensuring that the samples in the included articles were significant for the results. On the other hand, the limitations of our study lie in the fact that most available information focuses on Alzheimer’s disease and Parkinson’s disease as the primary types of dementia.

This study stands as one of the first systematic reviews aimed specifically at analyzing the effect of pharmacological therapy on sleep disorders related to dementia, addressing a crucial and timely question. The impact of our study lies in the discovery of the need to conduct more studies for the evaluation of all types of dementia and their management. Furthermore, it highlights the global lack of evidence regarding the effectiveness of pharmacological treatments for sleep disorders in patients with dementia.

Looking ahead, this review lays the groundwork for future studies to conduct a more comprehensive meta-analysis of the evidence, offering a more robust understanding of therapeutic effectiveness. We recommend that future studies include patient follow-up to evaluate whether beneficial effects are transient or can be established as foundational therapy in dementia patients and to assess whether improvements in the symptomatology of some patients are significant enough to be considered in the context of the disease.

## 5. Conclusions

The use of medication to treat sleep disorders in patients with dementia has shown promising results. Z-drugs, like Zopiclone and Eszopiclone, have been found to be more effective and have fewer side effects compared to benzodiazepines, which are currently the first-line pharmacological therapy used. This systematic review highlights the potential of Z-drugs to improve sleep architecture and dementia-related symptoms such as memory and language skills. It is essential to note that these medications should be personalized, weighing benefits against risks like sedation or falls. Improving sleep quality may contribute to a better quality of life for patients suffering from sleep disorders associated with dementia and pave the way for more effective management strategies in this vulnerable population. Future research should further clarify long-term safety and efficacy by comparing Z-drugs and benzodiazepines across dementia subtypes because the evidence is still limited and focused exclusively on Alzheimer’s disease.

## Figures and Tables

**Figure 1 ijms-26-05654-f001:**
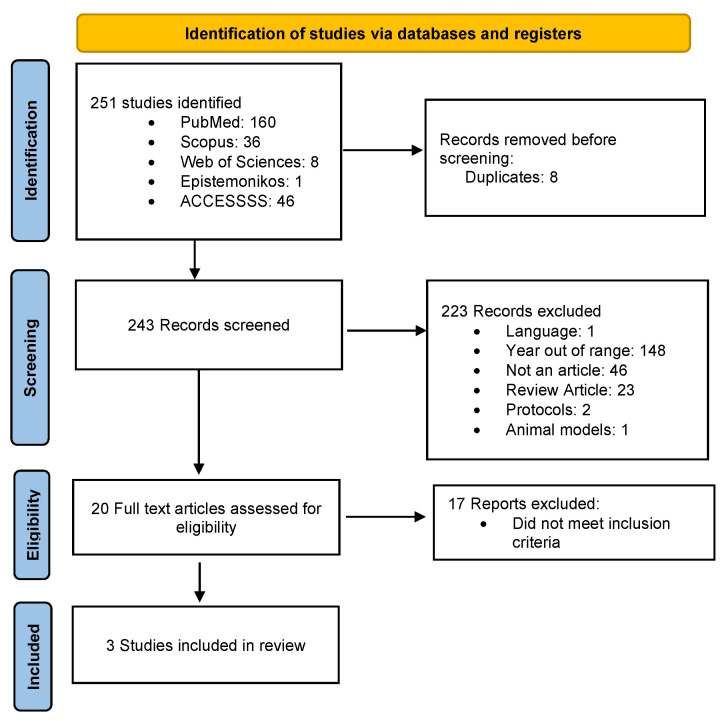
PRISMA flow diagram. This figure presents the results of the evaluation, analysis, and inclusion of articles for the systematic review. This is our own elaboration.

**Figure 2 ijms-26-05654-f002:**
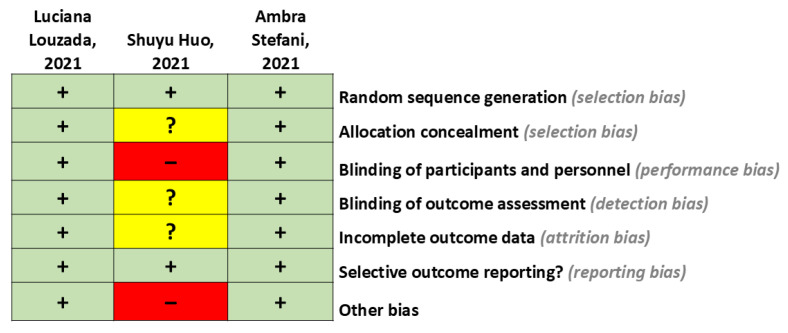
Risk of bias summary. This figure presents the review authors’ judgments regarding each risk of bias item for each included study. This is our own elaboration.

**Figure 3 ijms-26-05654-f003:**
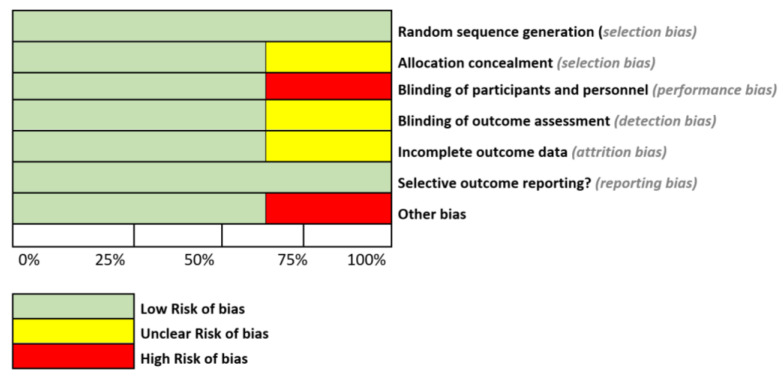
Risk of bias graph. The review authors’ judgments about each risk of bias item are presented as percentages across all included studies. This is our own elaboration.

**Table 1 ijms-26-05654-t001:** Summary of the characteristics of the analyzed studies.

Reference	Study Design	Follow-Up Duration	Sample	Objective Sleep Measure	Intervention/Comparison	Outcome
Stefani et al. (2021), USA [17]	Randomized, double-blind, placebo-controlled trial.	4 weeks	34 older adults (≥50 years) diagnosed with LBD or PDD and recurrent RBD (MMSE ≥ 18 or MoCA ≥ 12).	Video polysomnography (v PSG) was used to assess the number and duration of REM episodes and the nature of RBD. Movements and vocalizations in REM were classified according to the Innsbruck classification system.	*n* = 16: Nelotanserin 80 mg *n* = 18: Placebo	No significant difference was found between Nelotanserin and placebo in the change from baseline to week 4 in sleep behavior disorder events during simple/minor, simple/major, and complex REM phases per 10 min assessment frequency in REM sleep.
Louzada et al. (2022), Brazil [18]	Randomized, triple-blind, placebo-controlled trial	2 weeks	62 older adults (≥65 years) with probable late-onset Alzheimer’s dementia and insomnia (MMSE 0–24).	The main nocturnal sleep duration, duration of nocturnal awakenings, number of nocturnal awakenings, total daytime sleep duration, and number of daily naps were assessed.	*n* = 21: Zolpidem 10 mg/day *n* = 18: Zopiclone 7.5 mg/day *n* = 20: Placebo	Zopiclone led to an increase of 81 min in main nocturnal sleep duration; Zolpidem showed no significant difference. For the nocturnal awakening duration, Zopiclone and Zolpidem decreased awakenings by 26 and 21 min, respectively. Patients treated with Zopiclone had a reduced frequency and intensity of insomnia symptoms.
Huo et al. (2022), China [19]	Randomized trial controlled with alprazolam	4 weeks	*n* = 96 older adults (≥60 years) diagnosed with Alzheimer’s disease and sleep disorders per ICSD-3.	Pittsburgh Sleep Quality Index and EEG were used to record sleep progression and architecture.	*n* = 48: Eszopiclone 3 mg/day *n* = 48: Alprazolam 0.4 mg/day	Eszopiclone significantly improves sleep quality, benefits sleep progression and architecture, reduces mental symptoms such as fear or anxiety, significantly increases cognitive functions, and improves patients’ quality of life.

Table abbreviations: LBD (Lewy body dementia), PDD (Parkinson’s disease dementia), RBD (REM sleep behavior disorder), MMSE (Mini-Mental State Examination), MoCA (Montreal Cognitive Assessment), REM (rapid eye movement), MNSD (main nocturnal sleep duration), ICSD-3 (International Classification of Sleep Disorders), and EEG (electroencephalogram).

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
