# Peer review of "Dementia and Sleep Disorders: The Effects of Drug Therapy in a Systematic Review"

_ijms, 2025, doi:10.3390/ijms26125654_

Round 1
Reviewer 1 Report
Comments and Suggestions for Authors
Dear Authors,
this paper is well organized and presented
Author Response
Comment 1: Dear Authors, this paper is well organized and presented
Response 1: Thank you for your comment. The manuscript has been submitted for professional English language editing to ensure clarity and improve expression throughout the text. We will upload the final, corrected version upon resubmission.
Reviewer 2 Report
Comments and Suggestions for Authors
In this review about dementia and sleep disorders, the authors have missed telling the reader about some important dementing illnesses (especially DLB and PDD). These diseases are associated with some common sleep disorders (REM sleep disorder and OSA). OSA is also common in AD and VaD, so it needs to be mentioned in this review. The comments below provide some up-to-date references that should help to improve this review.
ABSTRACT (Methods): In the Methods section of the Abstract, it would be important to tell the reader which 5 databases were searched and how many authors performed the search (2) ...and over what timeline (5 years)...which 5 years?
ABSTRACT (Results): In the 1st sentence of the Results, please tell the reader about the # of papers included in the review and the # of patients included in the papers.
INTRO (lines 51-56): It is important to mention DLB and PDD, which both are associated with REM sleep disorder (see Stang et al Ann Neurol 2021....PMID 33155696).
INTRO (lines 57-59): When describing non-pharmacological therapies for sleep-related problems in dementia patients, it is good to mention light therapy for PDD (See Fifel et al Progr Neurobiol 2019....PMID 30658126). It is also good to mention in this section that CPAP has been suggested for treating OSA/sleep disordered breathing in patients with brain atrophy and cognitive decline (see Marchi et al Ann Neurol 2020....PMID 32220084). Another article that has discussed the role of OSA and its role in adverse brain health outcomes (and AD) is by Gottesman et al (see Stroke 2024.....PMID 38235581).
DISCUSSION (lines 170-71): This reads, "Dementia is considered one of the main conditions affecting the population today..." I would reword this to say, "Dementia is one of the most prevalent medical conditions affecting older adults today...." I would move right from there to state your hypothesis and your results: "In this review, we asked about drugs that were effective in treating sleep disorders in patients with dementia. We found that......" I would not go into detail in your discussion about the pathophysiology of AD, since that is not what this review was about. If you want to mention some studies about the role of sleep disorders in AD, then here is a good recent reference: Yoon et al Neurology 2023, PMID 37813585. Remember, focus on your data...that is what is important for a good review.
Author Response
Thank you very much for taking the time to review this manuscript. Please find the detailed responses below and the corresponding corrections highlighted purple in the re-submitted files.
Comment 1: In this review about dementia and sleep disorders, the authors have missed telling the reader about some important dementing illnesses (especially DLB and PDD). These diseases are associated with some common sleep disorders (REM sleep disorder and OSA). OSA is also common in AD and VaD, so it needs to be mentioned in this review. The comments below provide some up-to-date references that should help to improve this review.
Response 1: Thank you for your comment. We have added a sentence in the line 52 in Introduction mentioning other relevant types of dementia, including dementia with Lewy bodies (DLB) and frontotemporal dementia (FTD), although Alzheimer's disease, vascular dementia, and mixed dementia remain the most prevalent subtypes in Mexico, as reported in the reference.
Comment 2: ABSTRACT (Methods): In the Methods section of the Abstract, it would be important to tell the reader which 5 databases were searched and how many authors performed the search (2) ...and over what timeline (5 years)...which 5 years?
Response 2: Thank you for the suggestion. We have revised the Abstract to specify the five databases used, the number of authors involved in the search, and the time frame (line 29-31).
Comment 3: ABSTRACT (Results): In the 1st sentence of the Results, please tell the reader about the # of papers included in the review and the # of patients included in the papers.
Response 3: We have added the number of included studies. Regarding the total number of patients, since the studies varied in sample sizes and populations, we have provided an approximate total number derived from the included trials (line 35-36).
Comment 4: INTRO (lines 51-56): It is important to mention DLB and PDD, which both are associated with REM sleep disorder (see Stang et al Ann Neurol 2021....PMID 33155696).
Response 4: Thank you for your observation and for referencing a key study on the association between dementia types and REM sleep behavior disorder. In response, we revised the paragraph in the Introduction to explicitly include a general statement on the relationship between dementia and sleep disorders, such as insomnia, obstructive sleep apnea (OSA), and REM sleep behavior disorder (RBD), while noting that their occurrence varies depending on the type of dementia. Although dementia with Lewy bodies (DLB) and Parkinson’s disease dementia (PDD) are less prevalent in the Mexican population compared to Alzheimer’s disease, vascular dementia, and mixed types, we acknowledge their clinical importance and included a broader view of sleep disorders linked to different dementia subtypes in our updated manuscript (line 52 and 55-58).
Comment 5: INTRO (lines 57-59): When describing non-pharmacological therapies for sleep-related problems in dementia patients, it is good to mention light therapy for PDD (See Fifel et al Progr Neurobiol 2019....PMID 30658126). It is also good to mention in this section that CPAP has been suggested for treating OSA/sleep disordered breathing in patients with brain atrophy and cognitive decline (see Marchi et al Ann Neurol 2020....PMID 32220084). Another article that has discussed the role of OSA and its role in adverse brain health outcomes (and AD) is by Gottesman et al (see Stroke 2024.....PMID 38235581).
Response 5: Thank you for the suggestions. While light therapy and CPAP are relevant for specific types of sleep disorders (such as circadian rhythm disturbances and obstructive sleep apnea, respectively), our review addressed pharmacological interventions targeting general sleep disorders in dementia. We have maintained a general reference to non-pharmacological approaches—such as cognitive behavioral therapy and sleep hygiene—as first-line treatments, to preserve the focus and scope of our analysis. However, we acknowledge the relevance of these specific interventions and have added a sentence to clarify their role in particular sleep disorders, as suggested (LINE 64-66)
Comment 6: DISCUSSION (lines 170-71): This reads, "Dementia is considered one of the main conditions affecting the population today... "I would reword this to say, "Dementia is one of the most prevalent medical conditions affecting older adults today...." I would move right from there to state your hypothesis and your results: "In this review, we asked about drugs that were effective in treating sleep disorders in patients with dementia. We found that......" I would not go into detail in your discussion about the pathophysiology of AD, since that is not what this review was about. If you want to mention some studies about the role of sleep disorders in AD, then here is a good recent reference: Yoon et al Neurology 2023, PMID 37813585. Remember, focus on your data...that is what is important for a good review.
Response 6: Thank you for your observation. We revised the section and integrated a clear statement of our research question, as you suggested. Additionally, we slightly restructured the paragraph to make the aim of the review more explicit and to improve its flow (line 183 and 190-197).
Reviewer 3 Report
Comments and Suggestions for Authors
This review analyzes the effect of benzodiazepines and Z-drugs on sleep disorders in patients with dementia.
P7 line 233: The correct term is „REM sleep behavior disorder“ or RBD: Primary and secondary endpoints were manifestations of REM motor behaviors and their frequency in response to a 5HT receptor agonist, and not dementia per se. I do not really see where this study should meet the inclusion criteria, as only a few patients had been treated with clonazepam.
The conclusion seems exaggerated in stating that BDRA are superior to benzodiazepines with just one study showing this. I recommend to speak of results of a singular study. These results could certainly be supported by citing other studies in the discussion, which do not meet the inclusion criteria.
Author Response
Thank you very much for taking the time to review this manuscript. Please find the detailed responses below and the corresponding corrections highlighted in green in the re-submitted files.
Comment 1: This review analyzes the effect of benzodiazepines and Z-drugs on sleep disorders in patients with dementia. P7 line 233: The correct term is „REM sleep behavior disorder“ or RBD: Primary and secondary endpoints were manifestations of REM motor behaviors and their frequency in response to a 5HT receptor agonist, and not dementia per se. I do not really see where this study should meet the inclusion criteria, as only a few patients had been treated with clonazepam.
Response 1: Thank you for your observation. We appreciate your attention to detail and suggestion to use the correct terminology “REM sleep behavior disorder (RBD),” which we have now ensured is used consistently throughout the manuscript.
Regarding the inclusion of the study by Stefani et al., we respectfully clarify that, according to our predefined eligibility criteria (see Methods section), we considered not only benzodiazepines and Z-drugs but also innovative pharmacological interventions, due to their potential relevance in addressing sleep disorders in people with dementia. Specifically, criterion (3) under "Intervention" states: "use of benzodiazepines, Z-drugs, or any innovative medication."
Although the primary and secondary outcomes of the nelotanserin study focused on RBD manifestations, the study population consisted of patients diagnosed with dementia with Lewy bodies (DLB) or Parkinson’s disease dementia (PDD), which meets our inclusion criteria under the “Population” domain. Given that RBD is a common and clinically relevant sleep disorder in these neurodegenerative conditions, its treatment falls within the scope of our review. Moreover, while a few participants were receiving a stable dose of clonazepam, this was not the treatment under investigation and did not influence the study’s design or outcome interpretation. Instead, its use was allowed to maintain clinical stability during the trial.
To improve clarity for the reader, we have also revised the relevant paragraph in the Discussion section to better contextualize the inclusion of this study within the broader scope of our systematic review (line 258).
Comment 2: The conclusion seems exaggerated in stating that BDRA are superior to benzodiazepines with just one study showing this. I recommend to speak of results of a singular study. These results could certainly be supported by citing other studies in the discussion, which do not meet the inclusion criteria.
Response 2: Thank you for your comment. We believe there may be a misunderstanding regarding the term "BDRA," as it is not commonly used in the literature and was not mentioned in our manuscript. If you were referring to Z-drugs, we would like to clarify that two out of the three included studies evaluating Z-drugs (zopiclone, zolpidem and eszopiclone) reported improvements in sleep outcomes compared to alprazolam or placebo.
However, we agree that the current evidence is limited, and for that reason, our conclusion carefully states that Z-drugs "may have potential" and emphasizes the need for further studies to confirm their efficacy and safety in people with dementia. We appreciate your suggestion and have ensured the language in the conclusion remains cautious and grounded in the available evidence.
Reviewer 4 Report
Comments and Suggestions for Authors
The review “Dementia and sleep disorders: effect of drug therapy" discusses about the use of benzodiazepine vs Z-drugs on sleep disorder in patient with dementia. Authors provided the advantage of Z drugs in sleep disorder but did not discuss enough the disadvantage part of these drugs. Overall, the review covers the major area of its topic and aspects of use of non-benzodiazepine drugs improving sleep quality associated with dementia. However, the review requires a revision, including scientific writing style, improving sentences, and all the cited review articles should be rechecked. I think authors should also provide few lines about mechanistic of benzodiazepine and Z-drugs of action.
Lines 66-68, Sentence is not clear.
Lines 68-71, Sentence can be changed, "perfect hypnotherapy" can be replaced with different words.
Line 71, "Such a drug does not yet exist" this sentence can be improved. Citation number 8 is not in the English language.
Line 72, Wrong citation, the citation number 9 did not exactly study what authors mentioned here in a sentence.
Lines 76-78, Citation number 11 is a review, authors should also cite directly the research article which performed such studies.
Line 195, "more significant" can be replaced with "significant"
Lines 200-201, This sentence can be improved. And it lacks the citation.
Line 202-207, What do authors mean by "non-pharmacological treatments"? This paragraph is not clear.
Lines 232-24 are not relevant and can be removed.
Lines 267-269, "We can enhance the quality of life", This sentence can be improved.
Author Response
Thank you very much for taking the time to review this manuscript. Please find the detailed responses below and the corresponding corrections highlighted in yellow in the re-submitted file.
Comment 1: The review “Dementia and sleep disorders: effect of drug therapy" discusses about the use of benzodiazepine vs Z-drugs on sleep disorder in patient with dementia. Authors provided the advantage of Z drugs in sleep disorder but did not discuss enough the disadvantage part of these drugs. Overall, the review covers the major area of its topic and aspects of use of non-benzodiazepine drugs improving sleep quality associated with dementia. However, the review requires a revision, including scientific writing style, improving sentences, and all the cited review articles should be rechecked. I think authors should also provide few lines about mechanistic of benzodiazepine and Z-drugs of action.
Response 1: Thank you for this valuable observation. In response, we have expanded the discussion section (line 224) to include additional adverse effects of Z-drugs, such as behavioral changes and sex-related differences in metabolism, supported by relevant references. Additionally, we clarified the mechanism of action of both benzodiazepines and Z-drugs in the Introduction (line 82-92). The manuscript has also been revised for English language and style.
Comment 2: Lines 66-68, Sentence is not clear.
Response 2: The sentence has been revised to improve clarity and ensure better understanding (line 73-75).
Comment 3: Lines 68-71, Sentence can be changed, "perfect hypnotherapy" can be replaced with different words.
Response 3: We have rephrased the sentence to enhance its clarity and to convey the intended meaning more accurately (line 77-80).
Comment 4: Line 71, "Such a drug does not yet exist" this sentence can be improved. Citation number 8 is not in the English language.
Response 4: Thank you for pointing this out. The sentences have been rephrased (line 79). Also, we would like to clarify that the cited article is in Spanish because it is a publication from Madrid, Spain, which is used as a reference by the official Mexican Clinical Practice Guideline for sleep disorders. It supports the recommendation for pharmacological treatment in these disorders and is widely recognized in national clinical practice.
Comment 5: Line 72, Wrong citation, the citation number 9 did not exactly study what authors mentioned here in a sentence.
Response 5: We appreciate this observation. You are correct; reference 9 does not indicate use in patients with dementia but rather in older adults in general. We have corrected the paragraph accordingly (line 82).
Comment 6: Lines 76-78, Citation number 11 is a review, authors should also cite directly the research article which performed such studies.
Response 6: We understand your concern. In the paragraph where reference 11 is cited, we have left it as a general support, but more specific studies are added, cited and discussed later in the manuscript, particularly in the discussion section, such as those by Tao Guo, Richardson K., and Harbourt K., which directly address the effects of hypnotics in patients with dementia (line 230).
Comment 7: Line 195, "more significant" can be replaced with "significant"
Response7 : We have removed the word “more” to improve clarity and accuracy of the sentence (line 211).
Comment 8: Lines 200-201, This sentence can be improved. And it lacks the citation.
Response 8: We appreciate this suggestion. We have rewritten the sentence to enhance clarity and added citations of the specific studies reviewed (line 215).
Comment 9: Line 202-207, What do authors mean by "non-pharmacological treatments"? This paragraph is not clear.
Response 9: Non-pharmacological treatment refers to therapeutic interventions that do not involve medication or chemical substances. These may include physical or occupational therapy, dietary changes, lifestyle modifications, and psychological techniques such as relaxation therapy. In the context of sleep disorders, the most studied and recommended approach is cognitive behavioral therapy for insomnia (CBT-I), which has now been explicitly mentioned in the manuscript (line 218).
Comment 10: Lines 232-24 are not relevant and can be removed.
Response 10: Thank you. Although the comment was unclear in reference to the exact lines (either 232–234 or 222–224), we reviewed both sections and did not find inconsistencies that would affect the accuracy or clarity of the content. We opted to retain the current content, as it contributes relevant context to the discussion.
Comment 11: Lines 267-269, "We can enhance the quality of life", This sentence can be improved.
Response 11: Thank you for the feedback. The manuscript has been carefully revised for English language, grammar, and clarity throughout to ensure better readability and professionalism in academic writing (294-296).
Round 2
Reviewer 3 Report
Comments and Suggestions for Authors
The present form of the article is fine and all the reviewers comments have been responded to adequately